# The Role of Empathy in Alcohol Use of Bullying Perpetrators and Victims: Lower Personal Empathic Distress Makes Male Perpetrators of Bullying More Vulnerable to Alcohol Use

**DOI:** 10.3390/ijerph20136286

**Published:** 2023-07-03

**Authors:** Maren Prignitz, Tobias Banaschewski, Arun L. W. Bokde, Sylvane Desrivières, Antoine Grigis, Hugh Garavan, Penny Gowland, Andreas Heinz, Jean-Luc Martinot, Marie-Laure Paillère Martinot, Eric Artiges, Dimitri Papadopoulos Orfanos, Luise Poustka, Sarah Hohmann, Juliane H. Fröhner, Lauren Robinson, Michael N. Smolka, Henrik Walter, Jeanne M. Winterer, Robert Whelan, Gunter Schumann, Frauke Nees, Herta Flor, on behalf of the IMAGEN Consortium

**Affiliations:** 1Institute of Cognitive and Clinical Neuroscience, Central Institute of Mental Health, Medical Faculty Mannheim, Heidelberg University, Square J5, 68159 Mannheim, Germany; 2Department of Child and Adolescent Psychiatry and Psychotherapy, Central Institute of Mental Health, Medical Faculty Mannheim, Heidelberg University, Square J5, 68159 Mannheim, Germany; 3Discipline of Psychiatry, School of Medicine and Trinity College Institute of Neuroscience, Trinity College Dublin, Dublin 2, Ireland; 4Social, Genetic and Developmental Psychiatry Centre, Institute of Psychiatry, Psychology and Neuroscience, King’s College London, London SE5 8AF, UK; 5NeuroSpin, CEA, Université Paris-Saclay, F-91191 Gif-sur-Yvette, France; 6Departments of Psychiatry and Psychology, University of Vermont, Burlington, VT 05405, USA; 7Sir Peter Mansfield Imaging Centre School of Physics and Astronomy, University of Nottingham, University Park, Nottingham NG7 2QL, UK; 8Department of Psychiatry and Psychotherapy CCM, Charité—Universitätsmedizin Berlin, Corporate Member of Freie Universität Berlin, Humboldt-Universität zu Berlin, and Berlin Institute of Health, 10117 Berlin, Germany; 9Institut National de la Santé et de la Recherche Médicale, INSERM U 1299 “Trajectoires Développementales en Psychiatrie”, Université Paris-Saclay, CNRS, Ecole Normale Supérieure Paris-Saclay, Centre Borelli, F-91190 Gif-sur-Yvette, France; 10Department of Child and Adolescent Psychiatry, Pitié-Salpêtrière Hospital, AP-HP, Sorbonne Université, 75651 Paris, France; 11Psychiatry Department, EPS Barthélémy Durand, 91150 Etampes, France; 12Department of Child and Adolescent Psychiatry and Psychotherapy, University Medical Centre Göttingen, von-Siebold-Str. 5, 37075 Göttingen, Germany; 13Department of Child and Adolescent Psychiatry, Centre for Psychosocial Medicine, Heidelberg University, 69115 Heidelberg, Germany; 14Department of Psychiatry and Psychotherapy, Technische Universität Dresden, 01069 Dresden, Germany; 15Department of Psychological Medicine, Section for Eating Disorders, Institute of Psychiatry, Psychology and Neuroscience, King’s College London, London SE5 8AF, UK; 16Department of Education and Psychology, Freie Universität Berlin, 14195 Berlin, Germany; 17School of Psychology and Global Brain Health Institute, Trinity College Dublin, Dublin 2, Ireland; 18Centre for Population Neuroscience and Stratified Medicine (PONS), Department of Psychiatry and Neuroscience, Charité Universitätsmedizin Berlin, 10117 Berlin, Germany; 19Centre for Population Neuroscience and Precision Medicine (PONS), Institute for Science and Technology of Brain-Inspired Intelligence (ISTBI), Fudan University, Shanghai 200433, China; 20Institute of Medical Psychology and Medical Sociology, University Medical Center Schleswig Holstein, Kiel University, 24105 Kiel, Germany; 21Department of Psychology, School of Social Sciences, University of Mannheim, 68131 Mannheim, Germany

**Keywords:** bullying, alcohol, empathy, distress, adolescence

## Abstract

Bullying often results in negative coping in victims, including an increased consumption of alcohol. Recently, however, an increase in alcohol use has also been reported among perpetrators of bullying. The factors triggering this pattern are still unclear. We investigated the role of empathy in the interaction between bullying and alcohol use in an adolescent sample (IMAGEN) at age 13.97 (±0.53) years (baseline (BL), *N* = 2165, 50.9% female) and age 16.51 (±0.61) years (follow-up 1 (FU1), *N* = 1185, 54.9% female). General empathic distress served as a significant moderator of alcohol use in perpetrators (*F*_9, 493_ = 17.978, *p* < 0.01), which was specific for males and FU1. Male perpetrators, who are generally less sensitive to distress, might thus be more vulnerable to alcohol abuse.

## 1. Introduction

Bullying is a major issue in many domains of our social lives. Considering the definition of bullying from Olweus [1], someone is being bullied when he or she has to suffer over time from repeated, negative actions towards himself/herself by one or more other people. As a form of peer harassment at school [2], bullying has its main impact on adolescence and results in serious and diverse negative outcomes for victims. This includes internalizing problems, such as depression [3], together with an increase in alcohol use as a potential externalizing coping strategy [4]. However, studies also report on negative outcomes for perpetrators, mainly showing externalizing problems such as an increase in aggressive behavior [5,6], and also, similar to victims, an increase in alcohol use [7,8].

While the increase in alcohol use found in victims of bullying rather confirms expectations, it is still not clear why perpetrators often respond along those lines. One aspect that might come into play is emotional reactivity [9]. This develops in adolescence and is an important aspect of social challenges [10]. In this respect, emotional reactivity might either be beneficial for or hinder coping with negative events, including events of bullying. If victims’ coping fails due to strong emotional responses, they may develop or regress to alternative coping strategies such as alcohol use. This has already been shown in response to the experience of general negative events [11] as well as in response to highly stressful interpersonal situations [8,12,13].

If we consider the whole bullying process as a socially challenging situation, similar coping dynamics might also account for perpetrators. Even if perpetrators actively “create” such situations, those who are highly emotionally responsive might nevertheless also strongly react to these highly challenging events, similar to victims, and might fail in handling these situations [8,12,13]. This is in line with results from a meta-analysis by Kowalski et al. [14], who showed that perpetrators also tend to have higher levels of psychosocial distress and associated negative outcomes in bullying situations [15]. Moreover, if a perpetrator has higher general levels of personal empathic distress or empathic concern in intense interpersonal situations [16,17,18], he or she may also more strongly perceive any bullying situation as interpersonally challenging and stressful [12,15]. Thus, aside from emotional reactivity, for perpetrators, more general aspects of empathic distress or concern might specifically come into play. Empathy can be seen as a critical facet in regulating our social behavior [19], and indeed, it is defined as the ability to cognitively and affectively understand and share the emotions of others [20]. In this respect, general personal empathic distress, as an aspect of affective empathic processing, is supposed to reflect an individual disposition of a stronger self-focused and aversive emotional response when another person is under stress, which also includes the desire to reduce the accompanied stress in such a situation [16,18]. Thus, higher empathic personal distress can be seen as a “negative side” of empathic experience, which may be associated with other risk factors of heightened alcohol use [21]. This also fits with observations that, although perpetrators are mostly attributed with specific negative behavioral traits such as aggression [5], they still seem to have positive social skills [22] and are socially integrated into groups [23,24]. It can therefore be expected that being a perpetrator is more similar to a continuum of characteristics and influencing factors [25] in a situation-dependent manner rather than an all-or-none phenomenon, where handling in terms of “coping” with bullying situations might also be conceivable for perpetrators.

For empathy, it has also been shown that victims of bullying are characterized by significantly higher empathy compared to noninvolved participants [26]. Fabris et al. [15] postulate that this association is driven by the continuous experience of victimization, which may lead to better recognition of signals of other people’s suffering and where personal empathic distress might then come into play. If such associations exist, this might explain previous findings of increased alcohol use in both victims and perpetrators of bullying and thus explain alcohol use as a common medium for both bullying groups. Indeed, the disposition for empathic distress has been shown to be associated with a higher tendency to initiate negative coping behaviors, including alcohol use [27,28]. This tendency might even be increased in females, who usually score higher on measures of empathy than males [29,30]. A higher pattern of alcohol use among perpetrators and victims might therefore also be sex-specific, assuming that specifically female perpetrators react with higher alcohol use and that this might be driven by higher levels of empathy. The former has just been found in a very recent study [31]. Whether and how this is related to facets of empathy has, however, not been investigated.

In the present study, we, therefore, aimed to disentangle the so far under-represented role of empathy as a potential moderator in the association between bullying and alcohol use, compare both adolescent victims and perpetrators of bullying, and consider potential sex differences in a longitudinal dataset, which has so far rarely been used in research on bullying. We assumed perpetrators and victims would use more alcohol than noninvolved adolescents and have higher empathic personal distress, resulting in a further increase in this association (see Figure 1). Additionally, we expected to see differences driven by sex in these associations. Since internalizing and externalizing problems have been strongly observed in perpetrators and victims of bullying [12], we controlled for these characteristics.

## 2. Materials and Methods

### 2.1. Participants

Data for the current work were taken from the IMAGEN project, a longitudinal European multicenter study, with the aim of identifying genetic and neurobiological risk factors for developing psychological disorders in an ethnically homogenous but socioeconomically diverse sample [32]. Participants were assessed from adolescence to early adulthood along different waves (baseline (BL), follow-up 1 (FU1), follow-up 2, and follow-up 3) in eight cities in three European countries (the United Kingdom, France, and Germany). In the current work, we used data from BL and FU1. We only included participants, from whom complete datasets were available, leaving a total sample of 2165 participants at BL (mean age 13.98 ± 0.487 years, 50.9% female) and 1185 participants at FU1 (mean age 16.493 ± 0.643 years, 54.9% female). A detailed description of the present sample can be found in Table 1 and Table 2. Additionally, country-specific information on bullying distribution can be found in Appendix A.

### 2.2. Procedure

Participants underwent a large test battery, including neuropsychological measures (e.g., CANTAB), cognitive tasks (e.g., PALP), functional tasks (e.g., face task, monetary incentive delay task, stop signal task), structural magnetic resonance imaging tasks (volumetry, diffusion tensor imaging), and blood sampling for genotyping [32]. They also went through a large interview and questionnaire battery (e.g., on clinical characterization, personality, alcohol and drug use, and environmental factors) [32], assessed both at the study centers and online at home via the Psytools software [33]. For the present work, we selected questionnaire data on alcohol, empathy, bullying, and internalizing and externalizing problems.

### 2.3. Psychometric Assessments

Due to the multicenter and multilingual structure of the study, we used measures “based on three criteria: validation across [the] three languages, validation for use with adolescents, and suitability for electronic assessment” ([32], p. 1131) in the respective first language of the participants. Measures that did not meet these criteria were piloted and tested within the IMAGEN centers [32]. For the current work, we used three already validated questionnaires for empathy, alcohol use, internalizing/externalizing problems, and one questionnaire for bullying, which was piloted and tested within the IMAGEN study.

#### 2.3.1. Bullying

To measure bullying, a 12-item self-report questionnaire based on [34] was used at BL and FU1. Participants had to judge how often certain situations occurred in the past six months. Six items measure bullying perpetration (four items in peer context) and six items measure victimization (four items in peer context, see Appendix A). All items were answered on a 5-point scale from 0 = “None” to 4 = “Several times a week”. Cronbach’s alpha was acceptable, with *α* = 0.778 at BL and *α* = 0.693 at FU1.

For the present work, only bullying regarding peers was of interest. Therefore, we included the eight items asking for peer context in the categorization process and determined cutoffs in line with suggestions from Solberg and Olweus [35]: answers to items with “2 or 3 times a month” or more often. Participants were categorized into four bullying roles: pure perpetrators (responding to at least one perpetrator item), pure victims (responding to at least one victimization item), perpetrator-victims (responding to at least one perpetrator as well as one victimization item), and noninvolved students [35]. In our study, we are specifically interested in the differences between perpetrators and victims. The perpetrator-victim category, therefore, does not allow for a clear classification [12], nor does it add critical information, but rather noise to the data with respect to our research question. Therefore, we did not include this group in our analyses but decided to apply a three-group design with pure perpetrators, pure victims, and participants who were noninvolved in bullying as controls (see Table 1).

#### 2.3.2. Empathy

To measure trait empathy, we used the Interpersonal Reactivity Index (IRI by [36]) from FU1. The IRI has 28 items, evenly divided into four subscales: Perspective Taking (PT), Fantasy (F), Empathic Concern (EC), and Personal Distress (PD; see Appendix A), which reflect cognitive (PT and F) and affective aspects (EC and PD) of empathic processing [16]. Each item is rated on a 5-point scale from 1 = “does not describe me well” to 5 = “describes me very well”. The IRI is a well-validated and often-used instrument in empathy research. Cronbach’s alpha for each subscale was low (PT: *α* = 0.338, F: *α* = 0.292, EC: *α* = 0.115, PD: *α* = 0.378), but similar to other studies using the IRI in adolescent samples [37]. The IRI represents a trait-like disposition that is stable over time [16,38]. Based on previous studies [12,15,27], we were mainly interested in the subscale of Personal Distress. However, since it can also be argued that the different subscale-based dimensions interact and personal distress is not the sole factor in an empathic process, we also included the other subscales in our analyses. This allows a comparison and weighting of the empathic subfactors.

#### 2.3.3. Alcohol Use

To assess alcohol use, the Alcohol Use Disorder Identification Test (AUDIT by [39]) was used, both at BL and FU1. The AUDIT is a 10-item self-report measure to recognize risky and harmful alcohol use. Therefore, each item gets a score, and the sum score of all items is used to judge the alcohol use behavior. The AUDIT is a well-validated measurement and contains three conceptual axes (consumption pattern, dependency symptoms, and feature of harmful use) to capture a wide range of alcohol-associated problems. Cronbach’s alpha was good with *α* = 0.800 at BL and acceptable with *α* = 0.732 at FU1.

For alcohol use, we used AUDIT conceptual axis “consume pattern” as outcome [40]. This sum score of 3 items, also called “Quantity, frequency and hazardous use” (Q × F), reflects the intensity of drinking behavior, from no drinking (0) to heavy drinking (12).

#### 2.3.4. Internalizing and Externalizing Problems

To control for a potential role of internalizing and externalizing problems in the effect of bullying roles on alcohol use, we included the Strengths and Difficulties Questionnaire (SDQ by [41]), which was assessed at BL and FU1. The SDQ includes 25 items, equally divided into five subscales (emotional problems, conduct problems, hyperactivity/inattention, peer relationship problems, and prosocial behavior; see Appendix A). For the present work, we used the subscales externalizing problems (SDQ Extern, sum of emotional symptoms and peer relationship problems) and internalizing problems (SDQ Intern, sum of conduct problems and hyperactivity) pursuant to Goodman et al. [42].

### 2.4. Data Analysis

Analyses were conducted with SPSS for Windows, Version 25.0 [43], and a significance level of *p* < 0.05. Differences in sociodemographic data were explored using T-tests for independent samples. Based on previous studies, we first tested the main effect of bullying role (perpetrator vs. victim vs. noninvolved) on alcohol use. For this, we used (1) general linear models for baseline and follow-up 1 separately to also see the stability of this effect over time, with bullying role as 3-level and covariate sex as 2-level between-subject factors. In these models, SDQ Intern and SDQ Extern (for both models) and baseline alcohol use and baseline bullying role (for the FU1 model) were also entered as covariates. Then, we applied (2) general linear model for repeated measures, with time points (BL, FU1) as within-subject factors, bullying role at baseline as 3-level and covariate sex as 2-level between-subject factors, as well as covariates at baseline SDQ Intern and baseline SDQ Extern to test for effects on alcohol use as it increases over time. Post hoc tests were performed with univariate general linear model for baseline bullying role on alcohol use at follow-up 1 with bullying role at follow-up 1, SDQ Intern and SDQ Extern for baseline and follow-up 1, and sex as covariates, as well as with a difference score of alcohol use (FU1–BL) as dependent variable. To explore the moderating role of trait facets of empathy on the association between bullying role and alcohol use, both at baseline and follow-up 1, the SPSS plugin PROCESS macro version 3.4 [44] was used, and different models for each IRI subscale (PT, F, EC, and PD), with the covariates sex, SDQ Intern, and SDQ Extern for baseline alcohol use, and additional covariates follow-up 1 SDQ Intern, follow-up 1 SDQ Extern, baseline alcohol use, and baseline bullying role for follow-up 1 alcohol use, were calculated. In a post hoc analysis, we tested (1) if the effects are also present in the alcohol difference score (FU1 Q × F–BL Q × F) by 3-level-factor baseline bullying role and 3-level-factor follow-up 1 bullying role, with covariates sex, baseline alcohol use, and SDQ Intern and SDQ Extern, both at baseline and follow-up 1 and (2) if the bullying roles (3-level-between factor), each at baseline and follow-up 1, differ in internalizing and externalizing problems, using covariates sex for both models and additionally baseline bullying role and baseline SDQ Intern and baseline SDQ Extern, respectively, for follow-up 1 models.

## 3. Results

### 3.1. General Sample Information

We found no significant sex differences in age at baseline and follow-up 1 and alcohol use at baseline (all *p* > 0.05), but a significant difference in alcohol use at follow-up 1: males had higher scores than females (see Table 2). We also observed significant sex differences in all IRI subscales and for SDQ Intern, both at baseline and follow-up 1, with females having higher scores than males (see Table 2).

### 3.2. Main Effect of Bullying Role on Alcohol Use

#### 3.2.1. Baseline Bullying Role on Baseline Alcohol Use

There was a significant main effect of baseline bullying role on alcohol score at baseline (see Table 3). Pairwise comparisons showed that perpetrators (*M* = 1.736, *SD* = 2.117) had a significantly higher score than victims (*M* = 1.0, *SD* = 1.373) with *p* < 0.01 and noninvolved (*M* = 1.042, *SD* = 1.483) with *p* < 0.001. There was no significant difference between victims and noninvolved (*p* > 0.05) (see Figure 2), but a significant difference between females and males, as well as significant effects for the covariates SDQ Intern and SDQ Extern. Information on sex-separated models can be found in Appendix A.

#### 3.2.2. Follow-Up 1 Bullying Role on Follow-Up 1 Alcohol Use

There was a significant main effect of follow-up 1 bullying role on the alcohol score at follow-up 1. Pairwise comparisons showed that perpetrators (*M* = 4.500, *SD* = 2.918) had a significantly higher score than victims (*M* = 2.523, SD = 2.274) with *p* < 0.01 and noninvolved (*M* = 2.800, *SD* = 2.301) with *p* < 0.001. There was no significant difference between victims and noninvolved (*p* > 0.05) (see Figure 2), but a significant effect of the covariates sex, baseline alcohol score, follow-up 1 SDQ Intern, and follow-up 1 SDQ Extern (see Table 3). Information on sex-separated models can be found in Appendix A.

### 3.3. Alcohol Use Increases over Time

A significant main effect of time was found, indicating that participants had a higher score in alcohol use at follow-up 1 compared to baseline, and of bullying role, indicating perpetrators to have a higher alcohol score than victims (*p* < 0.001) and noninvolved (*p* < 0.001). Moreover, the interaction between time and sex reached significance, indicating a change in alcohol score in both sexes over time, and the main effects of the covariates sex, baseline SDQ Intern, and baseline SDQ Extern were found (all *p* < 0.01) (see Table 3).

#### Baseline Bullying Role on Follow-Up 1 Alcohol Use

There was no significant main effect of baseline bullying role on alcohol use at follow-up 1 (*p* > 0.05, see Figure 2), but a significant effect of covariates sex, baseline alcohol use, follow-up 1 bullying role, baseline SDQ Extern, follow-up 1 SDQ Intern, and follow-up 1 SDQ Extern (all *p* < 0.01, see Table 3). Due to the lack of a significant main effect of the baseline bullying role, we did not test the models separated for sex.

### 3.4. Moderation of the Relationship between Baseline Bullying Role and Baseline Alcohol Use by Empathy

We observed a significant interaction effect between baseline bullying role and the moderator for IRI F (*F*_6,1163_ = 8.854, *p* < 0.001, *R*^2^ = 0.044) (see Figure 3). Specifically, we found a significant main effect of IRI F, a significant interaction between baseline bullying role and F, and significant covariates sex, SDQ Extern, and SDQ Intern. For a detailed overview of the statistical values of all models, see Appendix A.

### 3.5. Moderation of the Relationship between Follow-Up 1 Bullying Role and Follow-Up 1 Alcohol Use by Empathy

We observed significant models for all empathy scores (see Table 4), however, mainly driven by significant effects of covariates sex, baseline alcohol use, SDQ Extern, and SDQ Intern at follow-up 1 (see Table 5). Only for the moderation model with IRI PD as moderator, we found a main effect of follow-up 1 bullying role in alcohol use in males, but not in females (see Figure 4), and a significant interaction between follow-up 1 bullying role and moderator IRI PD for males, but not females. Details on statistical values can be found in Table 4 and Table 5.

### 3.6. Post-Hoc Analysis with a Difference Score (FU1–BL) of Alcohol Use

There was a significant main effect of bullying role on the change in alcohol use (FU1 Q × F–BL Q × F, BL: *F*_2, 1537_ = 4.518, *p* < 0.05, η^2^ = 0.006; FU1: *F*_2, 1537_ = 7.826, *p* < 0.001, η^2^ = 0.010), with significant effects of the covariates baseline alcohol use (*F*_1, 1537_ = 97.281, *p* < 0.001, η^2^ = 0.060), follow-up 1 SDQ Extern (*F*_1, 1537_ = 43.269, *p* < 0.001, η^2^ = 0.027), follow-up 1 SDQ Intern (*F*_1, 1537_ = 23.576, *p* < 0.001, η^2^ = 0.015) and sex (*F*_1, 1537_ = 17.457, *p* < 0.001, η^2^ = 0.011). This association was not significantly moderated by empathy (all models *p* > 0.05).

### 3.7. Post-Hoc Analysis: Do Perpetrators, Victims, and Noninvolved Differ in Internalizing and Externalizing Problems?

There was a significant main effect of bullying in the SDQ subscale Internalizing Problems (BL: *F*_2, 2083_ = 129.803, *p* < 0.001, η^2^ = 0.111; FU1: *F*_2, 1556_ = 33.344, *p* < 0.001, η^2^ = 0.041), with significant effects of the covariate sex (BL: *F*_1, 2083_ = 97.710, *p* < 0.001, η^2^ = 0.045; FU1: *F*_1, 1556_ = 66.405, *p* < 0.001, η^2^ = 0.041) in both models. For follow-up 1 SDQ Intern, the covariate baseline SDQ Intern (*F*_1, 1556_ = 436.763, *p* < 0.001, η^2^ = 0.219), but not the baseline bullying role (*p* > 0.05), reached significance. Pairwise comparisons showed that victims (BL: *M* = 6.932, *SD* = 3.432, FU1: *M* = 8.253, *SD* =3.593) have significantly higher scores than perpetrators (BL: *M* = 4.141, *SD* = 2.990, FU1: *M* = 4.146, *SD* = 3.062) and noninvolved (BL: *M* = 4.048, *SD* = 2.613, FU1: *M* = 4.464, *SD* = 2.976) (all *p* < 0.001), whereas scores between perpetrators and noninvolved were not significantly different (all *p* > 0.05).

There was a significant main effect of bullying in the SDQ subscale Externalizing Problems (BL: *F*_2, 2083_ = 20.851, *p* < 0.001, η^2^ = 0.020, FU1: *F*_2, 1556_ = 3.446, *p* < 0.05, η^2^ = 0.004) for both models. Pairwise comparison showed that perpetrators (BL: *M* = 7.755, *SD* = 3.601; FU1: *M* = 6.195, *SD* = 3.558) have significantly higher scores than victims (BL: *M* = 6.441, *SD* = 3.198; FU1: *M* = 5.943, *SD* = 3.404—*p* < 0.01) and noninvolved (BL: *M* = 5.896, *SD* = 2.948; FU1: *M* = 5.105, *SD* = 2.971—*p* < 0.001), and that victims have significantly higher scores than noninvolved for baseline (*p* < 0.05), but not for follow-up 1 (all *p* > 0.05). The covariate sex did not reach significance in both models (both *p* > 0.05). Additionally, for follow-up 1 SDQ Extern, we found a significant effect of the covariate baseline SDQ Extern (*F*_1, 1556_ = 652.332, *p* < 0.001, η^2^ = 0.295), but not of the baseline bullying role (*p* > 0.05).

## 4. Discussion

Bullying behavior is associated with various individual risk outcomes for victims, but also perpetrators. While most of them, like depression and aggression, vary between victims and perpetrators, an increase in alcohol use has mostly been found in both [3,5,45]. Studies so far have tried to disentangle drinking motives as possible moderators of associations between different bullying roles and alcohol use and found enhancement and social motives to moderate alcohol use in perpetrators (in both sexes) and coping motives to moderate alcohol use in victims in males and at least partly also in females [8]. In this vein, we tested the role of trait empathy, in a large longitudinal sample of healthy adolescents.

We found an increase in alcohol use from BL to FU1 in our sample, with higher levels in males compared to females at FU1 [46,47,48] as well as higher levels of trait empathy scores in females compared to males [29,30]. This corroborates previous findings, and our sample might therefore be comparable with those from other studies in the distribution of our main variables.

When looking at the association between bullying role and alcohol use, perpetrators had not only higher levels of alcohol use than noninvolved, as previously reported [6], but also compared to victims. This was observed both at BL and FU1 and for the increase in alcohol use over time. Moreover, those associations were specific for males and not found in females, and also only in males a significant moderating effect of trait empathy, particularly of empathic personal distress at follow-up 1, was observed. In females, therefore other factors might play a role for the use of alcohol as a potential handling or coping mechanism in bullying. In male perpetrators, less empathic distressed ones had higher levels of alcohol use and alcohol use decreased with increasing empathy scores for this bullying role. This indicates that, in contrast to victims, where alcohol use usually serves as a coping mechanism in response to negative experiences of bullying [4], in perpetrators alcohol use may rather reflect a correlate of a “negative” trait with low empathic distress, as an aspect of affective emotional processing. This might be indicative of a potentially aggressive and/or psychopathic personality characteristic [18] and might result in less sensitivity to the social environment or social feedback. This is also supported by our results that perpetrators showed a significant increase in alcohol use over time, but without any effect of empathy. Previous studies already reported that males try to gain or maintain their status by showing aggressive behavior [23]. Moreover, adolescents with psychopathy traits score low on cognitive and affective empathy [49], show higher antisocial behavior [50] and positive associations to alcohol use [51]. This is related to our result that heightened fantasy, as a cognitive aspect of empathy, was associated with enhanced alcohol use in perpetrators at baseline. The IRI subscale Fantasy is defined as a tendency to identify oneself with fictional characters [36] rather than with real life situations. Therefore the heightened effect of fantasy on alcohol use in perpetrators might reflect the tendency to “create their own world”, which would go along with research showing perpetrators to consume media suiting their own beliefs and preferences, i.e., for aggression [52,53]. Taken together our results substantially amend these previous findings.

In contrast to previous studies [7], we found no significant difference in alcohol use between victims and noninvolved in our sample. Our sample of bullying victims might therefore rather be characterized by a stronger use of internalizing facets as coping strategies, indicated by our post hoc results and also found in previous research [3,54]. We also found no significant effect of trait empathy on alcohol use in bullying victims, indicating that empathy might indeed serve as a perpetrator-specific trait in the context of alcohol use. Moreover, different forms of bullying may be differentially be affected by trait empathy, which could also explain the sex-specific effects: for example, physical aggression is more strongly used by males, whereas relational aggression has more often been observed for females [6,13] and also bullying intensity is often stronger when bullying is performed by males vs. females [55]. It might therefore be assumed that bullying intensity has an effect on changing the use of alcohol to handle or cope with bullying. This also indicates the presence of subgroups of perpetrators and victims with different empathic abilities and therefore different patterns of alcohol use. However, these aspects still need to be tested explicitly in future studies.

Some limitations of the current study should be mentioned. We did not control for possible confounders like general drinking motives [8], socioeconomic status variables like educational level or household income [56,57,58,59,60], cultural background [61], personality facets [62,63], popularity [64,65] or other forms of peer victimization [66]. Taking all these contributing factors into account, future studies might benefit from a comprehensive approach and complex analyses like structural equation modeling.

Moreover, we had a rather large dropout rate with respect to the bullying data. This resulted in rather small bullying groups with an unequal size, and thus also a smaller overlap between BL and FU1 (see Table 1). However, they also map previously reported lower prevalence rates for bullying behavior in adolescence [35]. Given that we have longitudinal data, which are still rather rare, our data can thus still add valid information in the context of bullying, empathy and alcohol use during adolescence. Finally, the lower number of female participants in the perpetrator group should be considered when interpreting sex effects. Nevertheless, we mainly included sex as a covariate in our models, due to previous research suggesting differences in bullying behavior [31], empathy [30], and alcohol use [47] between sexes. Future studies on the association of bullying and alcohol use should therefore look at a younger age, where bullying behavior occurs more frequently and the lasting effects of (state) empathy might be further followed in longitudinal designs.

Additionally, self-report on bullying behavior could be seen as an undesirable behavior, and research showed only a small overlap between self-report and peer nomination in bullying behavior [67]. Furthermore, the self-reported trait empathic score, which showed low internal consistency in the current work, could be empowered by using other empathy questionnaires (e.g., Questionnaire of Cognitive and Affective Empathy [68] or Basic Empathy Scale [69]) and additional state-like measures, both on a behavioral as well as physiological level [70]. This could also provide further information in terms of sex-specific differences in empathy based on self-reports [71]. Nevertheless, at the start of the IMAGEN project, the IRI was the best-validated and most used questionnaire in empathy research, and due to the longitudinal assessment strategy, it was kept within the project.

Moreover, for the present study, we excluded the perpetrator-victims group as it does not allow a clear classification in terms of our specific research question [12]. Additionally, the size of this group in our sample was rather small, which does not allow any add-on analyses. However, in terms of alcohol abuse, one could also argue that specifically this double role serves an interesting aspect, which is also underlined by some previous work showing that perpetrator-victims might be even more at risk for negative outcomes associated with bullying [72]. Future research on empathy and bullying behavior might therefore benefit from including this high-risk group, if it is meaningful with respect to the research question and if the sample size is large enough. Along with this, future research should also disentangle how stressful and negative perpetrators experience bullying situations they are involved in [73] and if this (co-)modulates their alcohol use in dealing with the bullying situation.

Finally, our results can also be discussed in the context of bullying and related negative outcome prevention in adolescence [74,75,76,77], where empathy enhancement might be a promising target to decrease the risk of alcohol use in a bullying situation and potentially also decrease bullying itself. Along those lines, promising interventions might be the Olweus Bullying Prevention Program [78] and mindfulness-based stress reduction programs [79], which may also function as prevention approaches.

## 5. Conclusions

The current paper aimed to disentangle whether empathy has an impact on the alcohol use behavior of adolescents when they are perpetrators or victims of bullying. We found perpetrators, but not victims, to use more alcohol than noninvolved adolescents. In males, this association was moderated by personal empathic distress; perpetrators with decreasing personal empathic distress showed an increase in alcohol use. With decreasing empathic distress, behaviors, such as aggression or psychopathic characteristics, often described in perpetrators, might prevail over any empathic feeling for the victims` situation. As a consequence, this might provoke more inappropriate and harmful behaviors, which could be reflected in an increase in alcohol use. On the other hand, our findings suggest that females and bullying victims may use more internalizing strategies to cope with bullying experiences. Therefore, empathy enhancement might be relevant in preventing bullying and its negative consequences in adolescents.

## Figures and Tables

**Figure 1 ijerph-20-06286-f001:**
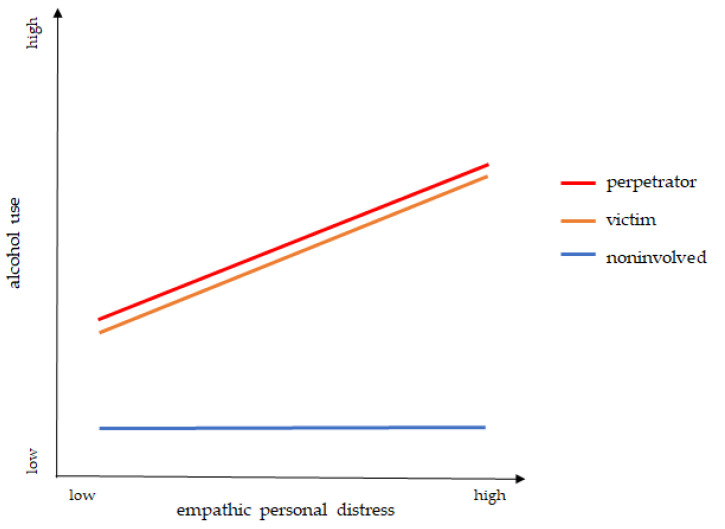
Graphical illustration of the hypotheses: (1.) perpetrators and victims are assumed to consume more alcohol than noninvolved individuals and (2.) higher empathic personal distress is assumed to increase this association.

**Figure 2 ijerph-20-06286-f002:**
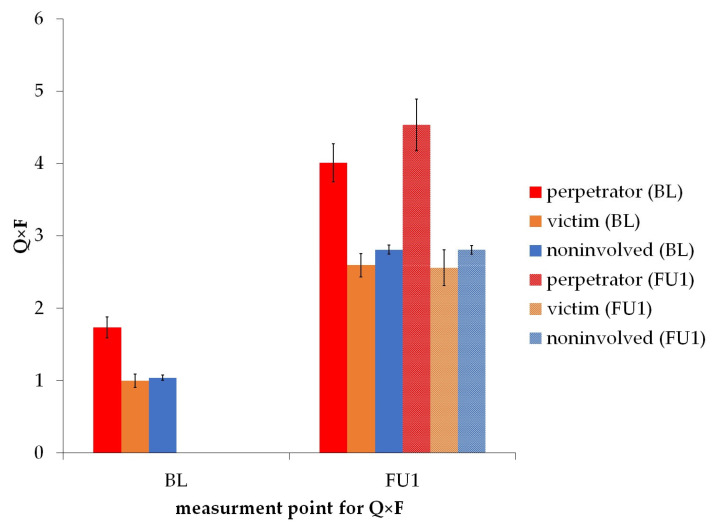
Alcohol Use Disorder Identification Test “Quantity × Frequency” (Q × F) score separated by measurement point. Structure of bars represent measurement point of bullying with BL = baseline and FU1 = follow-up 1, error bar represents standard error.

**Figure 3 ijerph-20-06286-f003:**
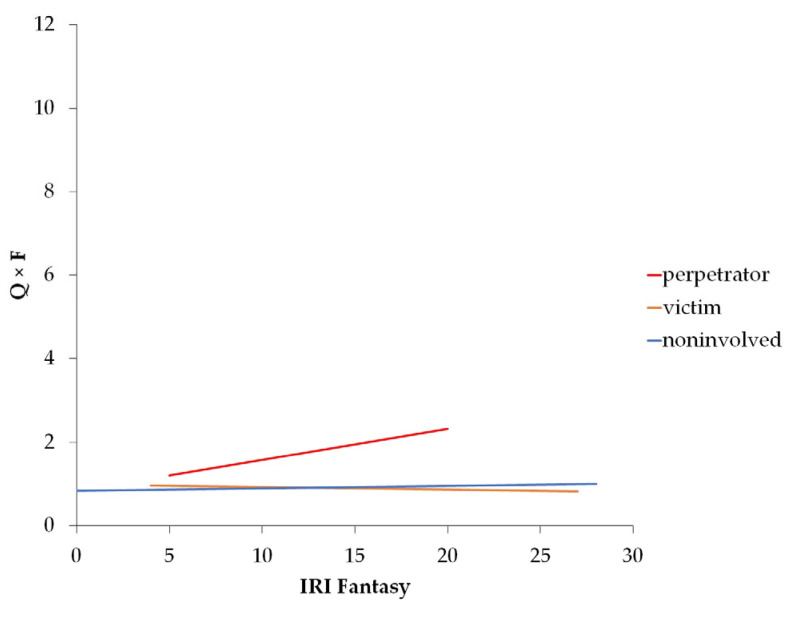
Moderation effect of IRI subscale Fantasy on the association between bullying role and Alcohol Use Disorder Identification Test Quantity × Frequency score (Q × F) both measured at baseline.

**Figure 4 ijerph-20-06286-f004:**
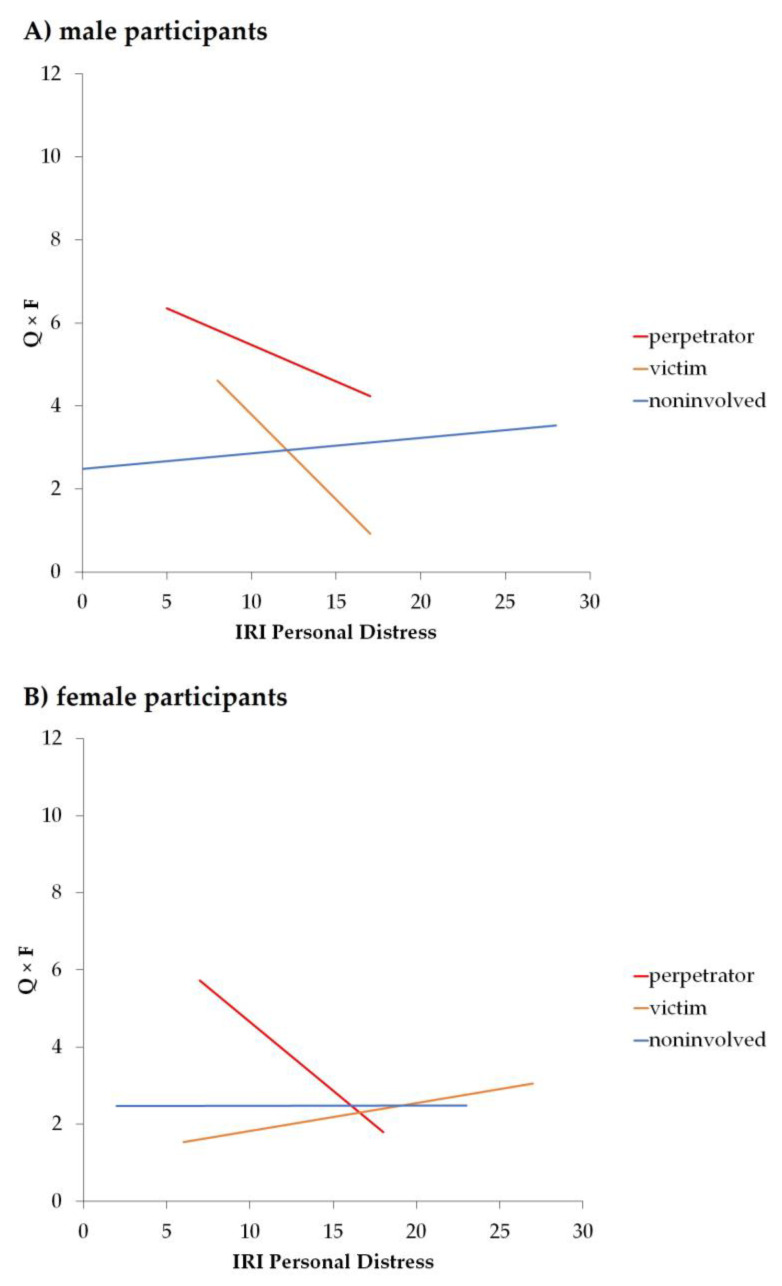
Moderation effect of Interpersonal Reactivity Index subscale of Personal Distress on the association between follow-up 1 bullying role and Alcohol Use Disorder Identification Test Quantity × Frequency score (Q × F), separated by sex: (**A**) male participants, (**B**) female participants.

**Table 1 ijerph-20-06286-t001:** Sample distribution.

			Total		Male		Female	
			*N*	(%)	*N*	(%)	*N*	(%)
BL	Participants		2165		1062	(49.1)	1103	(50.9)
	Language	English	853	(39.4)	431	(40.6)	422	(38.3)
		French	261	(12.1)	130	(12.2)	131	(11.9)
		German	1050	(48.5)	500	(47.1)	550	(49.9)
	Bullying role	Perpetrator	106	(4.9)	77	(7.3)	29	(2.6)
		Victim	276	(12.7)	115	(10.8)	161	(14.6)
		Perp.-Victims	72	(3.3)	55	(5.2)	17	(1.5)
		Noninvolved	1711	(79.0)	815	(76.7)	896	(81.2)
FU1	Participants		1185		533	(45.0)	650	(54.9)
	Bullying role	Perpetrator	32	(2.7)	17	(3.2)	15	(2.3)
		Victim	63	(5.3)	16	(3.0)	46	(7.1)
		Perp.-Victims	15	(1.3)	10	(1.9)	5	(0.8)
		Noninvolved	1075	(90.7)	490	(91.9)	584	(89.8)
OL	Bullying role	Perpetrator	8	(25.0) *	5	(29.4) *	3	(20.0) *
		Victim	28	(44.4) *	9	(56.3) *	19	(41.3) *
		Perp.-Victims	3	(20.0) *	2	(20.0) *	1	(20.0) *
		Noninvolved	873	(81.3) *	383	(78.3) *	489	(83.7) *

*Note*. BL = baseline, FU1 = follow-up 1, OL = Overlap bullying cases between BL and FU1, *N* = total number of cases, % = percentage of all cases, * = percentage of cases compared to groups at FU1.

**Table 2 ijerph-20-06286-t002:** Sample description.

			Total		Male		Female		Sex Differences	
			*M*	(*SD*)	*M*	(*SD*)	*M*	(*SD*)	*t*	(*df*)
BL	Age		13.973	(0.532)	13.981	(0.502)	13.964	(0.560)	0.715	(2221)
	Q × F		1.091	(1.559)	1.074	(1.576)	1.115	(1.548)	−0.611	(2177)
	SDQ	I	4.490	(2.951)	3.879	(2.817)	5.083	(2.959)	−9.766	(2194) ***
		E	6.156	(3.115)	6.287	(3.255)	6.027	(2.969)	1.953	(2194)
FU1	Age		16.511	(0.611)	16.509	(0.633)	16.514	(0.591)	-0.154	(1650)
	Q × F		2.863	(2.330)	3.166	(2.528)	2.581	(2.088)	5.211	(1695) ***
	SDQ	I	4.729	(3.149)	3.844	(2.848)	5.562	(3.194)	−11.695	(1690) ***
		E	5.290	(3.089)	5.224	(3.124)	5.355	(3.058)	−0.871	(1690)
	IRI	PT	15.218	(3.033)	14.900	(3.142)	15.478	(2.918)	−3.319	(1214) ***
		EC	14.760	(2.367)	14.544	(2.560)	14.931	(2.189)	−2.801	(1214) **
		PD	13.487	(3.128)	12.750	(3.082)	14.087	(3.042)	−7.58	(1213) ***
		F	15.113	(3.442)	14.504	(3.706)	15.608	(3.124)	−5.55	(1216) ***

*Note*. BL = Baseline; FU1 = follow-up 1; Q × F = Alcohol Use Disorder Identification Test quantity × frequency sub score; SDQ = Strengths and Sifficulties Questionnaire; I = subscale Internalizing Problems; E = subscale Externalizing Problems; IRI = Interpersonal Reactivity Index; PT = subscale Perspective Taking; EC = subscale Empathic Concern; PD = subscale Personal Distress; F = subscale Fantasy; *M* = mean; *SD* = standard deviation; *t* = t value; *df* = degrees of freedom; ** *p* < 0.01; *** *p* < 0.001.

**Table 3 ijerph-20-06286-t003:** Statistical values of all general linear models for the effects of bullying on alcohol use.

Dependent Variable	Covariates/*Predictor*	*F*	(*df*1,*df*2)	*p*	ƞ^2^
BL Q × F	sex	**5.587**	**(1, 2073)**	**0.018**	**0.003**
	BL SDQ I	**10.617**	**(1, 2073)**	**0.001**	**0.005**
	BL SDQ E	**92.418**	**(1, 2073)**	**<0.001**	**0.043**
	*BL BR*	**6.557**	**(2, 2073)**	**0.001**	**0.006**
FU1 Q × F	sex	**18.687**	**(1, 1542)**	**<0.001**	**0.012**
	BL SDQ I	0.000	(1, 1542)	0.998	0.000
	BL SDQ E	0.162	(1, 1542)	0.687	0.000
	BL BR	0.381	(1, 1542)	0.537	0.000
	BL Q × F	**297.321**	**(1, 1542)**	**<0.001**	**0.162**
	FU1 SDQ I	**22.206**	**(1, 1542)**	**<0.001**	**0.014**
	FU1 SDQ E	**41.455**	**(1, 1542)**	**<0.001**	**0.026**
	*FU1 BR*	**7.686**	**(2, 1542)**	**<0.001**	**0.010**
Q × F over Time	time	**165.785**	**(1, 1617)**	**<0.001**	**0.093**
	time × sex	**37.328**	**(1, 1617)**	**<0.001**	**0.023**
	time × BL SDQ I	1.581	(1, 1617)	0.209	0.001
	time × BL SDQ E	2.866	(1, 1617)	0.091	0.002
	time × BL BR	0.685	(2, 1617)	0.504	0.001
	sex	3.329	(1, 1617)	0.068	0.002
	BL SDQ I	**9.825**	**(1, 1617)**	**0.002**	**0.006**
	BL SDQ E	**56.914**	**(1, 1617)**	**<0.001**	**0.034**
	*BL BR*	**9.588**	**(2, 1617)**	**<0.001**	**0.012**
FU1 Q × F	sex	**17.895**	**(1, 1542)**	**<0.001**	**0.011**
	BL SDQ I	0.103	(1, 1542)	0.748	0.000
	BL SDQ E	**45.192**	**(1, 1542)**	**<0.001**	**0.028**
	FU1 BR	**9.946**	**(1, 1542)**	**0.002**	**0.006**
	BL Q × F	**292.428**	**(1, 1542)**	**<0.001**	**0.159**
	FU1 SDQ I	**26.472**	**(1, 1542)**	**<0.001**	**0.017**
	FU1 SDQ E	**45.192**	**(1, 1542)**	**<0.001**	**0.028**
	*BL BR*	2.514	(2, 1542)	0.081	0.003

*Note.* BL = baseline, FU1 = follow-up 1, Q × F = Alcohol Use Disorder Identification Test Quantity × Frequency subscore, SDQ I = Strengths and Difficulties Questionnaire subscale “Internalizing Problems”, SDQ E = Strengths and Difficulties Questionnaire subscale “Externalizing Problems”, BR = bullying role, main predictor of each model is italic, bold marked values are significant at *p* < 0.05 or below.

**Table 4 ijerph-20-06286-t004:** General model information of separated moderator models for each IRI subscale as moderators on the association between FU1 bullying role and FU1 alcohol use, separated by sex.

	Total			Male			Female		
Model	*F*	*p*	*R* ^2^	*F*	*p*	*R* ^2^	*F*	*p*	*R* ^2^
PT	32.512	<0.001	0.225	16.841	<0.001	0.235	18.427	<0.001	0.211
F	32.693	<0.001	0.226	16.980	<0.001	0.236	18.089	<0.001	0.208
EC	32.280	<0.001	0.223	16.935	<0.001	0.236	18.021	<0.001	0.208
PD	33.061	<0.001	0.228	17.978	<0.001	0.247	17.910	<0.001	0.210

*Note.* “Model” refers to the applied moderator, PT = Perspective Taking, F = Fantasy, EC = Empathic Concern, PD = Personal Distress.

**Table 5 ijerph-20-06286-t005:** Detailed description of separated moderator models for each IRI subscale as moderators on the association between FU1 bullying role and FU1 alcohol use, separated by sex.

		Total			Male			Female		
Model	*Predictors*/Covariates	*b*	*t*	*p*	*b*	*t*	*p*	*b*	*t*	*p*
PT	constant	**5.637**	**3.235**	**0.001**	3.369	1.161	0.246	**5.834**	**2.685**	**0.007**
	*BR*	−0.654	−1.467	0.143	−0.136	−0.186	0.853	−0.995	−1.780	0.076
	*PT*	−0.119	−1.082	0.279	0.045	0.236	0.813	−0.236	−1.761	0.079
	*BR* × *PT*	0.025	0.869	0.385	−0.016	−0.321	0.748	0.053	1.524	0.128
	BL BR	−0.054	−0.756	0.500	−0.106	−0.995	0.320	0.010	0.101	0.920
	BL Q × F	**0.647**	**14.431**	**<0.001**	**0.766**	**10.131**	**<0.001**	**0.546**	**10.098**	**<0.001**
	BL SDQ E	−0.027	−1.038	0.299	−0.058	−1.415	0.158	0.001	0.041	0.968
	BL SDQ I	0.019	0.713	0.476	0.035	0.798	0.425	−0.001	−0.044	0.965
	FU1 SDQ E	**0.157**	**6.231**	**<0.001**	**0.182**	**4.480**	**<0.001**	**0.138**	**4.340**	**<0.001**
	FU1 SDQ I	**−0.113**	**−4.504**	**<0.001**	**−0.169**	**−3.792**	**<0.001**	**−0.066**	**−2.249**	**0.025**
	sex	**−0.440**	**−3.427**	**0.001**						
F	constant	**4.928**	**2.676**	**0.007**	**5.375**	1.744	0.082	2.052	0.903	0.367
	*BR*	−0.667	−1.441	0.150	−0.796	−1.025	0.306	−0.283	−0.494	0.622
	*F*	−0.066	−0.594	0.553	−0.085	−0.444	0.657	0.003	0.025	0.980
	*BR* × *F*	0.025	0.879	0.380	0.028	0.571	0.568	0.008	0.232	0.816
	BL BR	−0.045	0.632	0.528	−0.104	−0.980	0.328	0.025	0.260	0.795
	BL Q × F	**0.648**	**14.474**	**<0.001**	**0.757**	**10.025**	**<0.001**	**0.552**	**10.147**	**<0.001**
	BL SDQ E	−0.025	−0.941	0.347	−0.056	−1.371	0.171	0.002	0.050	0.960
	BL SDQ I	0.014	0.544	0.587	0.032	0.738	0.461	−0.004	−0.121	0.904
	FU1 SDQ E	**0.155**	**6.160**	**<0.001**	**0.180**	**4.467**	**<0.001**	**0.143**	**4.497**	**<0.001**
	FU1 SDQ I	**−0.116**	**−4.591**	**<0.001**	**−0.173**	**−3.877**	**<0.001**	**−0.074**	**−2.493**	**0.013**
	sex	**−0.480**	**−3.732**	**<0.001**						
EC	constant	3.512	1.469	0.142	7.121	1.681	0.093	−1.185	−0.419	0.676
	*BR*	−0.259	−0.424	0.672	−1.241	−1.148	0.252	0.659	0.906	0.365
	*EC*	0.024	0.150	0.881	−0.209	−0.728	0.467	0.217	1.193	0.234
	*BR* × *EC*	−0.001	−0.028	0.978	0.060	0.815	0.416	−0.054	−1.138	0.256
	BL BR	−0.049	−0.686	0.493	−0.110	−1.027	0.305	0.031	0.315	0.753
	BL Q × F	**0.647**	**14.413**	**<0.001**	**0.760**	**10.060**	**<0.001**	**0.545**	**10.065**	**<0.001**
	BL SDQ E	−0.028	−1.051	0.294	−0.057	−1.385	0.167	−0.002	−0.052	0.959
	BL SDQ I	0.014	0.538	0.590	0.033	0.750	0.454	−0.005	−0.152	0.879
	FU1 SDQ E	**0.156**	**6.209**	**<0.001**	**0.180**	**4.457**	**<0.001**	**0.147**	**4.628**	**<0.001**
	FU1 SDQ I	**−0.113**	**−4.519**	**<0.001**	**−0.172**	**−3.880**	**<0.001**	**−0.070**	**−2.372**	**0.018**
	sex	**−0.457**	**−3.566**	**<0.001**						
PD	constant	**7.304**	**4.951**	**<0.001**	**10.060**	**4.110**	**<0.001**	3.425	1.794	0.073
	*BR*	**−1.218**	**−3.232**	**0.001**	**−2.035**	**−3.256**	**0.001**	−0.442	−0.967	0.334
	*PD*	**−0.252**	**−2.443**	**0.015**	**−0.477**	**−2.529**	**0.012**	−0.092	−0.737	0.462
	*BR × PD*	**0.069**	**2.585**	**0.010**	**0.133**	**2.752**	**0.006**	0.022	0.660	0.510
	BL BR	−0.054	−0.759	0.448	−0.114	−1.080	0.281	0.015	0.153	0.879
	BL Q × F	**0.649**	**14.503**	**<0.001**	**0.767**	**10.232**	**<0.001**	**0.54**	**10.049**	**<0** **.001**
	BL SDQ E	−0.207	−1.027	0.305	−0.060	−1.477	0.140	−0.001	−0.019	0.985
	BL SDQ I	0.016	0.591	0.554	0.038	0.864	0.388	−0.001	−0.027	0.978
	FU1 SDQ E	**0.154**	**6.131**	**<0.001**	**0.182**	**4.516**	**<0.001**	**0.144**	**4.550**	**<0** **.001**
	FU1 SDQ I	**−0.110**	**−4.374**	**<0.001**	**−0.179**	**−4.047**	**<0.001**	**−0.066**	**−2.228**	**0** **.026**
	sex	**−0.463**	**−3.589**	**<0.001**						

*Note.* “Model” refers to the applied moderator, PT = Perspective Taking, F = Fantasy, EC = Empathic Concern, PD = Personal Distress, BR = bullying role at follow-up 1, BL BR = bullying role at baseline, BL Q × F = Alcohol Use Disorder Identification Test Quantity × Frequency score at baseline, BL SDQ E = Strengths and Difficulties Questionnaire subscale “Externalizing Problems” at baseline, BL SDQ I = Strengths and Difficulties Questionnaire subscale “Internalizing Problems” at baseline, FU1 SDQ E = Strengths and Difficulties Questionnaire subscale “Externalizing Problems” at follow-up 1, FU1 SDQ I = Strengths and Difficulties Questionnaire subscale “Internalizing Problems” at follow-up 1, main predictors are written in italic, bold marked coefficients are significant at *p* < 0.05 or below.

## Data Availability

Ethical restrictions to protect participant confidentiality prevent us from making anonymized study data publicly available. This also refers to the analysis/experimental code, and any other digital materials, where participant-related anonymized information are also included. Readers seeking access to the study data and materials should contact the corresponding author based on a formal collaboration agreement. The data and materials will be released to requestors after approval of this formal collaboration agreement by the local ethics committees.

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
