# Peer review of "The Role of Empathy in Alcohol Use of Bullying Perpetrators and Victims: Lower Personal Empathic Distress Makes Male Perpetrators of Bullying More Vulnerable to Alcohol Use"

_ijerph, 2023, doi:10.3390/ijerph20136286_

Round 1

Reviewer 1 Report

Dear Authors,

Thank you for the opportunity to review the manuscript entitled “Lower personal empathic distress makes male perpetrators of bullying more vulnerable for alcohol use”. Please find my comments below.

-       Last sentence of the abstract needs editing

- The intro needs editing

-       How measures were validated after translation to the mother tongue language?

-       Not sure why not included the bully-victim group in the analysis especially the literature showed that this group experience more psychological problems than the pure victim and pure perpetrator groups and therefore, may consume alcohol more than the latter ones. I saw that this has been explained in the limitations section.

-       How many items were the “consume pattern of alcohol subscale?

-       In the introduction, the role of empathy among bullying victimization has been emphasized more than those for perpetrators, therefore, I suggest to change the title of the paper to include both. Reading only the title shows that only perpetrators of bullying were included in the study.

-       It is not clear what the authors are trying to postulate, i.e., is it lower or higher empathy moderates the relationship between bullying role and alcohol use? It is increased or decreased personal distress moderate such relationship? I suggest to make the argument up to the point and write research hypothesis

-       There should be conceptualization about empathy especially the IRE. The IRE consists of 4 scales and every two subscales targets one dimension of empathy; empathic concern and personal distress relates to emotional empathy, while perspective taking and fantasy represent the cognitive dimension of empathy. It is important to differentiate between cognitive and emotional empathy as some individuals may score higher on one and deficient on the other

-       Since the paper especially the discussion section focused on the role of personal distress as a moderator, conceptualization of this concept is important. PD as an index of emotional empathy has been found to behave differently than PT, F, and EC. Recently, it has been found that personal distress represents the negative side of emotional empathy and could block empathic interaction instead of enhancing it. Contradicting results were reported in the study in Supplementary (S6) and Table 4 in the manuscript. In the supplementary S6, PD has a negative moderation role between bullying role at follow up and alcohol consumption and the moderation was not significant, but a significant and positive role the Table.

-       The significant role of fantasy as a main effect and moderator was not explained in the discussion. Furthermore, the result BR x F was not presented in Table 4 as well but present in the Supplementary (S6)

Overall, the argument and direction of the relationship between the variables in addition to revisiting the analysis and discussion section is needed.

Thank you...

Reviewer 2 Report

The paper titled "Lower Personal Empathic Distress Makes Male Perpetrators of Bullying More Vulnerable for Alcohol Use " addresses an important current topic: Bullying among adolescents and emerging adults. A strength of the article is that it analyzes different bullying groups (i.e., victims, perpetrators), which are compared to noninvolved in bullying practices adolescents. Another important strength is the large sample size from different cultural contexts. The literature review was thorough and up to date. The research methods were sound, and the results definitely contributed greatly to the literature on bullying. I found this article to be well-written and presented.

This study has the potential to contribute to expanding knowledge about the relationship between alcohol use and empathic distress among adolescents involved in bullying. I recommend accepting this paper in its current form. However, I would like to suggest one major and one minor points.

I found the description of the IMAGEN project to be very brief. Please expand on it and add more on the period of data collection, study design and sample procedures. Importantly, please discuss how have you addressed the threats to the internal validity of the study?

Minor points

* It might be useful for future meta-analyses to see more context-specific (country-specific) comparisons of bullying. Please include a table with this information.

I congratulate the authors for this work, which is thorough and exhaustive, and considerably adds to the knowledge.

Reviewer 3 Report

I have carefully reviewed this manuscript and below is my decision.

-The article title and abstract should be changed.

- The topic is quite interesting, however, the explanation on the originality of the study is insufficient. This paper, needs to highlight clearly the originality of the study.

- How does the paper contribute to the extant literature on the subject?

-I would suggest adding to the literature and referencing it within the introduction and discussion as well.  There are studies that have examined intimate partner violence.

1) https://doi.org/10.1007/s11356-021-12849-2

2) https://doi.org/10.1080/14659891.2020.1846811

  It can be published after corrections are made.

Reviewer 4 Report

The current study is from an angle that is less likely to be explored (the mental health of the Bully as well as the victim). The longitudinal nature of the study is also a positive. Below I have some suggestions to enhance the manuscript. The major issue is the low internal consistency for the IRI. I have made a suggestion to run a factor analysis on this scale to see if there is a more stable solution.

Abstract:

Minor: The lasts sentence of the abstract seems incomplete (e.g., should it be “Male perpetrators who are…..)

Introduction:

1.      For inclusivity I would not use the pronoun “he” in a general definition of bullying. I understand that the study focuses on males, but it seems out of place in the general definition.

2.      The definition of Empathy seems a bit disconnected from the first part of the paragraph. I would like to see a clearer connection drawn between the two parts of this paragraph (lines 93-94)

3.      It would be great to see a model outlining the predictions of your study. Including all the predictors, interactions, outcomes etc. This would improve the clarity of the introduction.

Methods:

1.      This is a minor issue but some of the language used in this section is a little casual “e.g., mother longue (i.e., first language), took about 15 minutes (approximately 15 minutes)”.

2.      Major: The very low internal consistently of the IRI is a bit concerning. I would suggest running an exploratory factor analysis, to investigate which items are perhaps not working in this scale format. You might find the items actually load onto different factors, and some items do not work at all. I worry with such low internal consistency it is difficult to make any firm conclusions based on the IRI.

3.      I found the later section of the methods a little hard to follow with the overuse of acronyms.

Results:

1.      Again, I found the results a little hard to follow with the overuse of acronyms.

2.      Minor: I think Figure 3 could benefit from a panel-based approach. I.e., either have 2 panels split by gender or 3 panels split by group (i.e., victim, noninvolved….). At the moment it looks a little busy.

Discussion:

1.      The discussion could benefit from a clinical implication section. How might empathy be targeted in therapy?

Round 2

Reviewer 3 Report

It can be published.

Reviewer 4 Report

I think the authors have responded to my comments and those of the other reviewer well.